

# Screening of salt tolerance of maize (*Zea mays* L.) lines using membership function value and GGE biplot analysis

Huijuan Tian, Hong Liu, Dan Zhang, Mengting Hu, Fulai Zhang, Shuqi Ding and Kaizhi Yang

College of Agriculture, Tarim University, Alar, China
Key Laboratory of Genetic Improvement and Efficient Production for Specialty Crops in Arid Southern Xinjiang of Xinjiang Corps, Alar, China

Corresponding author
Dan Zhang, zdzkytd@163.com

## ABSTRACT

Soil salinization is a widely recognized global environmental concern that has a significant impact on the sustainable development of agriculture at a global scale. Maize, a major crop that contributes to the global agricultural economy, is particularly vulnerable to the adverse effects of salt stress, which can hinder its growth and development from germination to the seedling stage. This study aimed to screen highly salt-tolerant maize varieties by using four NaCl concentrations of 0, 60, 120, and 180 mMol/L. Various agronomic traits and physiological and biochemical indices associated with salt tolerance were measured, and salt tolerance was evaluated using principal component analysis, membership function method, and GGE biplot analysis. A total of 41 local maize varieties were assessed based on their D values. The results show that stem thickness, germ length, radicle length, leaf area, germination rate, germination index, salt tolerance index, and seed vigor all decreased as salt concentration increased, while electrical conductivity and salt injury index increased with the concentration of saline solution. Under the stress of 120 mMol/L and 180 mMol/L NaCl, changes in antioxidant enzymes occurred, reflecting the physiological response mechanisms of maize under salt stress. Principal component analysis identified six major components including germination vigor, peroxidase (POD), plant height, embryo length, SPAD chlorophyll and proline (PRO) factors. After calculating the comprehensive index (D value) of each variety's performance in different environments using principal component analysis and the membership function method, a GGE biplot analysis was conducted to identify maize varieties with good salt tolerance stability: Qun Ce 888, You Qi 909, Ping An 1523, Xin Nong 008, Xinyu 66, and Hong Xin 990, as well as varieties with poor salt tolerance: Feng Tian 14, Xi Meng 668, Ji Xing 218, Gan Xin 2818, Hu Xin 712, and Heng Yu 369. Furthermore, it was determined that a 120 mMol/L NaCl concentration was suitable for screening maize varieties during germination and seedling stages. This study further confirmed the reliability of GGE biplot analysis in germplasm selection, expanded the genetic resources of salt-tolerant maize, and provided theoretical references and germplasm utilization for the introduction of maize in saline-alkali areas. These research findings contribute to a better understanding of maize salt tolerance and promote its cultivation in challenging environments.

## INTRODUCTION

Soil salinization has seriously affected the sustainable development of global agriculture, resulting in a significant reduction in crop yield and quality worldwide (*Jiang et al., 2020*; *Neupane et al., 2022*). The area of saline land in the world is as high as 954 million hectares, and the area of saline land in China reaches 37 million hectares, accounting for 4.9% of the arable land (*Li et al., 2020*; *Luo et al., 2018*). The physiology and biochemistry of plants growing in saline soils were affected, affecting all plant stages from seed germination to nutritional and reproductive development (*Hasanuzzaman, Nahar & Fujita, 2013*). Maize (*Zea mays* L.) is a widely cultivated multi-purpose crop for food, feed, and energy, which plays an important role in the development of the world economy (*Ramazan et al., 2023*). It is a moderately saline crop with low salt tolerance and is severely affected by salt stress at the seedling stage. Screening of maize varieties with good salt tolerance lays the foundation for the development of salt-tolerant germplasm suitable for cultivation in saline and alkaline soils, improvement of saline and alkaline soils, promotion of sustainable development of agriculture and the breeding, production and utilization of salt germplasm resources.

Salt stress negatively affects various phenotypic traits, physiological functions, and biochemical indices of plants, especially during reproductive stages like germination and seedling growth (*Munns, 2005*; *Song et al., 2008*). Previous research has found that salt damage significantly reduces maize yield stability (*Chen et al., 2019*). It leads to decreased seed germination, hinders plant growth rate and yield, and impairs the plant's ability to uptake water (*Ghatak et al., 2016*; *Foolad, Hyman & Lin, 1999*; *Carpýcý, Celýk & Bayram, 2009*). Salt-tolerant varieties exhibit higher peroxidase activity under salt-stress conditions (*Wang et al., 2017a*). After salt stress, maize leaf conductivity and malondialdehyde content were significantly increased with peroxidative damage to the cytoplasmic membrane (*Rubio et al., 2009*), but plant proline (PRO) levels were also significantly increased to alleviate the damage and enable plant growth (*Ben Ahmed et al., 2010*). The salt tolerance indicators of different fertility periods and treatments were also distinct. Various indicators have been used to assess the salt tolerance of maize, such as germination rate, germination potential, germ length, radicle length, seedling condition, plant height, and change in dry weight (*Zhao et al., 2014*; *Qi et al., 2022*; *Fu, Gao & Wang, 2009*).

Numerous methods have been employed for screening salt tolerance in plants. Many previous studies have utilized the affiliation function method, which calculates the affiliation function value and the D value based on weights, to evaluate overall plant performance under abiotic stresses, such as drought tolerance, salt tolerance, and cold tolerance (*Yu et al., 2018*; *Zhang & Zhao, 2016*). Other analytical methods, including flux and stratification analyses, as well as GGE biplot analyses, have been used to evaluate different species. However, previous studies did not combine D-values from the affiliation function method with GGE biplots analysis to evaluate cultivar resistance (*Deng et al., 2021*; *Wang et al., 2017c*). GGE biplots analysis is a valuable tool for categorizing environments, assessing genotypic sorting, determining differentiation and representativeness of environments (*Oladosu et al., 2017*), analyzing crop yield and stability (*Yu et al., 2020*;

*Flores et al., 2013*), identifying ideal environments for crop growth (*Choudhary et al., 2019*), screening and evaluating germplasm resources (*Oliveira et al., 2019*; *Yan et al., 2011*). This method graphically illustrated the relationship between cultivars, identification indicators, or evaluation methods through auxiliary lines, providing accurate insights into cultivar performance in different environments and appropriate metrics for identification or evaluation (*Yan et al., 2000*). Previous researchers used GGE biplot analysis to study heat tolerance in wheat by analysing late yield and multipoint trials (*Gupta et al., 2023*; *Saeidnia, Taherian & Nazeri, 2023*). Similarly, this analysis is commonly used for adaptation and stability analyses in crops such as sorghum (*Yan, 2002*), sugarcane (*Todd et al., 2018*), groundnut (*Muhammad et al., 2020*), and soybean (*Silva et al., 2022*). Other analytical methods, including flux and stratification analyses, as well as GGE biplot analyses, have been used to assess different species. However, previous studies did not combine the D-values from the affiliation function method with GGE biplot analysis to evaluate cultivar resistance.

While significant research has been conducted on the salt tolerance mechanisms of maize, most studies have focused on individual stages without considering other reproductive phases. Few studies have utilized GGE biplot analysis to investigate maize germination and seedling stages under salt stress. In this study, previous work was expanded by simulating various agronomic traits and physiological indicators related to salt tolerance in maize through different NaCl concentrations. NaCl concentrations of 0, 60, 120, and 180 mMol/L were used to create different salt stress conditions. By combining multiple analytical methods and employing the affiliation function method and GGE biplots, the accuracy and reliability of our evaluations were enhanced. In this study, the salt tolerance and stability of 41 maize varieties were comprehensively evaluated, and the suitable NaCl concentration and key evaluation indexes were determined, which provided valuable guidance for the selection and promotion of maize varieties in saline-alkali areas.

## MATERIALS & METHODS

### Plant materials
For this trial, 41 maize varieties popularized in production were selected as test materials, and the specific material information is shown in Table 1.

### Experimental design
#### Germination test for seeds indoors
Germination experiments were conducted in a light-illuminated thermostatic incubator with a day/night temperature of 25 °C (day/night) and 75% humidity at the study took place at the Key Laboratory of Genetic Improvement and Efficient Production for Specialty Crops in Arid Southern Xinjiang of Xinjiang Corps in April-May 2022. The experiment was a two-factor randomised block design. The experiment was set up with a total of four concentration gradients of 0 (CK), 60, 120, and 180 mmol/L NaCl concentrations and each treatment was replicated three times. The Petri dish was first sterilised with 70% alcohol, two layers of filter paper were placed in the dish, and then the seeds were placed evenly in the dish. Incubation was carried out with four concentrations of NaCl concentration, 0, 60,

**Table 1  Maize materials tested.**

| Number | Name | Source | Number | Name | Source |
|---|---|---|---|---|---|
| 1 | Hong Xing 990 | Jilin Hongxing Seed Industry Co | 22 | Zheng Dan 958 | Henan Academy of Agricultural Sciences |
| 2 | Ping An 1523 | Jilin Ping'an Seed Industry Co | 23 | HengYu 369 | Jilin Hengyu Seed Industry Co. |
| 3 | Hu Xin 712 | Huludao Agriculture Co | 24 | Xiang He 9918 | Changtu Dingsheng Agricultural Co. |
| 4 | Zhong Xing 618 | Inner Mongolia Seedstar Co | 25 | Xing Nong No.1 | Zhongyan Seed Industry Co. |
| 5 | Xi Meng No.6 | Inner Mongolia Simon Co | 26 | Bi Xiang 809 | Beijing Huanong Weiye Seed Industry Co. |
| 6 | Hong Xing 528 | Jilin Hongxing Seed Industry Co | 27 | Jin Fengjie 607 | Zhengzhou Fengjie Seed Co. |
| 7 | Zeng Yu 157 | Jilin Hongxing Seed Industry Co | 28 | Ji Nongyu 309 | College of Agriculture, Jilin Agricultural University |
| 8 | Wo Feng 188 | Shanxi Wodafeng Co | 29 | Xin Yu 81 | Xinjiang Pioneer Weiye Seed Co. |
| 9 | Yu He 536 | Henan Yuyu Seed Industry Co | 30 | Xin Yu 24 | Xinjiang Xinshi Seed Co. |
| 10 | Wo Feng No.9 | Shanxi Vodafone Co. | 31 | Deng Hai 3672 | Shandong Denghai Seed Industry Co. |
| 11 | Nong Fu 99 | Inner Mongolia Zhongnong Co | 32 | Xin Yu 66 | Urumqi Shengyang Agricultural Company |
| 12 | Xuan He No.8 | Xuanhui Agriculture Co | 33 | Xin Nong 008 | Inner Mongolia Lanhai Xinnong Agricultural Company |
| 13 | Xi Meng 668 | Inner Mongolia Simon Co | 34 | Xian Yu 335 | Tieling Pioneer Seed Co. |
| 14 | Lin Yu 1339 | Yunnan Linpeng Agriculture Co | 35 | Feng Tian 14 | Chifeng Fengtian Science and Technology Seed Industry Co. |
| 15 | Yuan Yuan No.1 | Weishan Jiyuan Agricultural Co. | 36 | Fu Yu 109 | Zhongyan Seed Industry Co. |
| 16 | Qun Ce 888 | Sichuan Qunze Seed Industry Co | 37 | Xi Meng 3358 | Inner Mongolia Simon Co. |
| 17 | Shan Ning No.23 | Ningxia Jinyu Seed Co | 38 | Ji Xing 218 | Jilin Xingnong Seed Industry Co. |
| 18 | You Qi 909 | Jilin Hongxiang Seed Industry Co | 39 | San Meng 9599 | Changtu Zewei Agricultural Science Research Institute |
| 19 | Jin An 588 | Inner Mongolia Jin'ai Ailite Co | 40 | Xi Meng 208 | Inner Mongolia Simon Co. |
| 20 | Gan Xin 2818 | Gansu Province Wuwei Agricultural Research Institute | 41 | Hu Xin 338 | Huludao Agriculture Co. |
| 21 | Wu Gu 568 | Gansu Wugu Seed Industry Co | | | |

120, and 180 mmol/L, and the solution in each Petri dish soaked more than 1/3 of the seeds. Each day, the Petri dish was filled with the appropriate amount of NaCl concentration at the appropriate concentration. The seed germination was recorded at 16:00 h each day. The seed germination potential was determined after 4 d of treatment, and the germination rate was determined after 7 d of treatment. The germ length and radicle length were measured. In addition, the relative germination potential, relative germination rate, salt tolerance index, and salt injury index were calculated. The mentioned measurements and calculations were made with reference to *Liu et al. (2015)*.

## Physiological seedling measurements in outdoor experiment

The study was conducted at the Agricultural Experimental Station of Tarim University (40°32′N, 81°18′E) from May to July 2022. The average temperature from May to July was 32 °C/14 °C, and the average monthly rainfall was 0.16 mm. 25 × 25 cm seedling pots were used. The experiment was a two-factor randomised block design. The growing medium was fine sandy soil with a soil weight of 4.5 kg. The PH of fine sandy soil was around 6.8 ± 0.5. The sowing depth of each variety of maize seeds was about three cm, 10 grains were sown in each pot, and 120 grains with uniform size were selected. When seedlings grew to three leaves and one heart, they were treated with nutrient solution containing 1/4 concentration of Hoagland nutrient solution and different concentrations of salt solution, the nutrient solution applied concentration referred to *Jia et al. (2022)*. The treatment concentrations

were 0 (control, Controls are those without the addition of salt concentration treatments, referred to as CK in this paper), 60, 120, and 180 mMol/L NaCl concentration, and each treatment was repeated three times. The treatments were carried out for 7 days (*Wang et al., 2017a*), after which plant height, stem thickness, leaf area, chlorophyll SPAD, and dry and fresh weights were measured, and root crown ratio and water content were calculated. The distance from the base of the plant to the highest point was determined as plant height using a straightedge (*Wang et al., 2017a*), and the stem thickness was measured as the average diameter of the plant by the vernier caliper (*Meng et al., 2021*). For each replication, five maize plants were taken and the second and third leaves of each plant were measured at the same position using a SPAD502 instrument, and the measured values of the five plants were averaged to obtain a fixed chlorophyll SPAD value for each replication (*Jia et al., 2022*). The aboveground and belowground parts of the seedlings were separated, placed at 105 °C for 30 min, and dried at 80 °C until constant weight. Then the dry weight of the biomass was weighed, and the root-crown ratio (root-crown ratio = aboveground dry weight/underground dry weight) was calculated (*Wang et al., 2017a*). The tip of the second leaf was cut into 5–6 small pieces and the fresh weight (FW) was recorded. Then the leaves were immersed in 10 mL deionised water in a test tube at room temperature for 24 h. After 24 h, the leaves were wiped with a paper towel and the swelling weight (TW) was recorded. The dry weight was recorded after drying in an oven at 70 °C for 72. The relative water content was (FW-DW)/(TW-DW) × 100 (*Ghoulam, Foursy & Farès, 2002*). Five maize plants were selected for each replication and the 2nd and 3rd expanded leaves of each plant were taken and measured by a LI3100 benchtop leaf area instrument from LI-COR, Lincoln, NE, USA. Determination of conductivity (*Lutts, Kinet & Bouharmont, 1996*), MDA (*Li, 2000*), SOD (*Giannopolitis & Ries, 1977*), POD, PRO. The POD and PRO were determined by the PRO and POD test kits of Suzhou Keming Biotechnology Company, Suzhou, China. 0.1 g fresh sample in a mortar was weighed, one mL extraction solution was added, and then it was homogenised in an ice bath; after that, the sample was extracted by shaking in a water bath at 95 °C for 10 min; 10,000 g was centrifuged at 25 °C for 10 min, and the supernatant was taken. After cooling, the absorbance was measured by a spectrophotometer, and the PRO content was calculated. Later, 0.1 g fresh sample in a mortar was weighed and homogenised in an ice bath. 8,000 g was centrifuged at 4 °C for 10 min, the supernatant was taken and placed on the ice to be measured. The instrument used was a spectrophotometer, and the POD activity was calculated.

## Calculation formula of identification index

Based on the salt tolerance coefficient (salt tolerance coefficient = mean value of salt stress index/mean value of control index) of each identification index of the test varieties, 41 maize varieties were comprehensively evaluated for salt tolerance by using the subordinate function method in fuzzy mathematics (*Gao et al., 2020*), the calculation formula referred to the Gao et al. (*Rao et al., 1997*). and the calculation formula was as follows

Coefficient of variation (CV) = (standard deviation SD/mean value) × 100%.

The formula for calculating the membership function value is as follows: Xi is the salt tolerance coefficient of each test material based on the identification index i, Xi max and
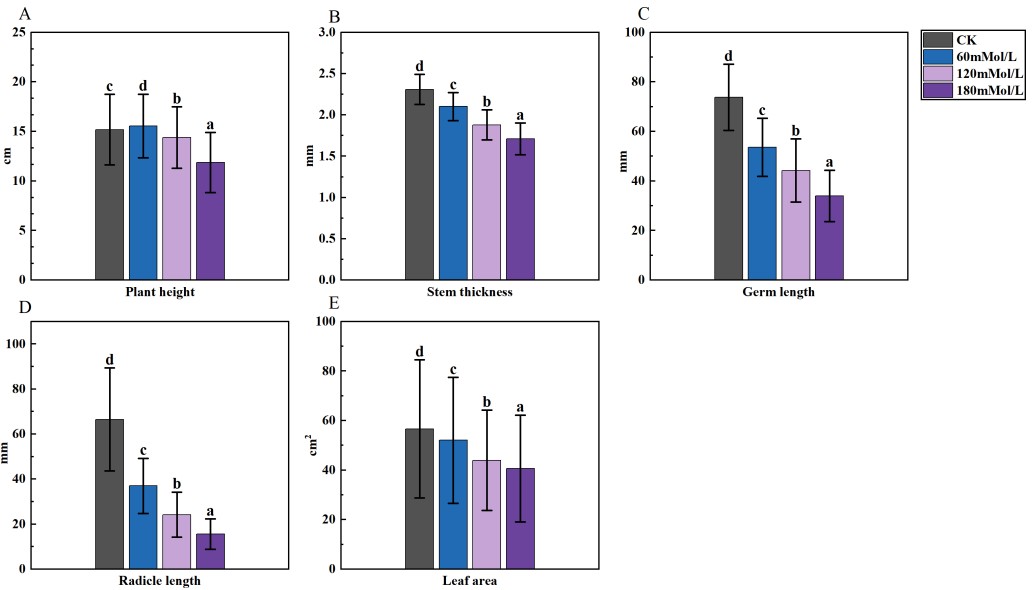

**Figure 1** **Analysis of difference in various morphological indicators under salinity stress.** (A–D) Markers of significance of differences at 0.05 level.

Xi min are the maximum and minimum values of Xi in the test materials, respectively, and U(Xi) is the membership function value of each test material Xi.

Inverse membership function formula: $U(Xi) = 1 - (Xi - Ximin)(Ximax - Ximin)$

The weighting formula: $Wi = CVi / \Sigma_{I=1}^{n} CVi (i = 1, 2, 3, \ldots, n)$. CVi is the coefficient of variation of each test material U(Xi) and Wi is the ratio of CVi to the total variation.

Weighted membership function value formula: $D = \Sigma_{I=1}^{n} [U(Xi) \cdot Wi] (i = 1, 2, 3, \ldots, n)$. U(Xi) is the value of the membership function of each test material Xi and Wi is the weight of each identification index.

## Data analysis

IBM SPSS Statistics 25 software was used to analyze the variance of examined traits of 41 maize varieties. The results of the ANOVA analysis were plotted in Figs. 1–3 with Origin 2022 software. Origin 2022 software was used to analyze the correlation traits of 41 maize varieties, as shown in Fig. 4. IBM SPSS Statistics 25 software was used to analyze the principal component of 41 maize varieties. The D-values corresponding to the six principal components of the 41 extracted maize varieties were analysed and plotted in Figs. 5–8 using GenStat 9.2 software, partly based on a previous study (*Kumar et al., 2023*). Due to data processing, the data were analysed using Genstat software so that 0, 60, 120 and 180 mMol/L NaCl concentrations were denoted as C1, C2, C3 and C4, respectively, as shown in Fig. 5-figure supplement 8. ANOVA analysis was performed on the selected 12 maize varieties using Origin 2022 software and plotted in Fig. 9.

ANOVA and regression analysis were performed with a significant confidence interval of $P < 0.05$, and the Pearson test was used for correlation analysis.

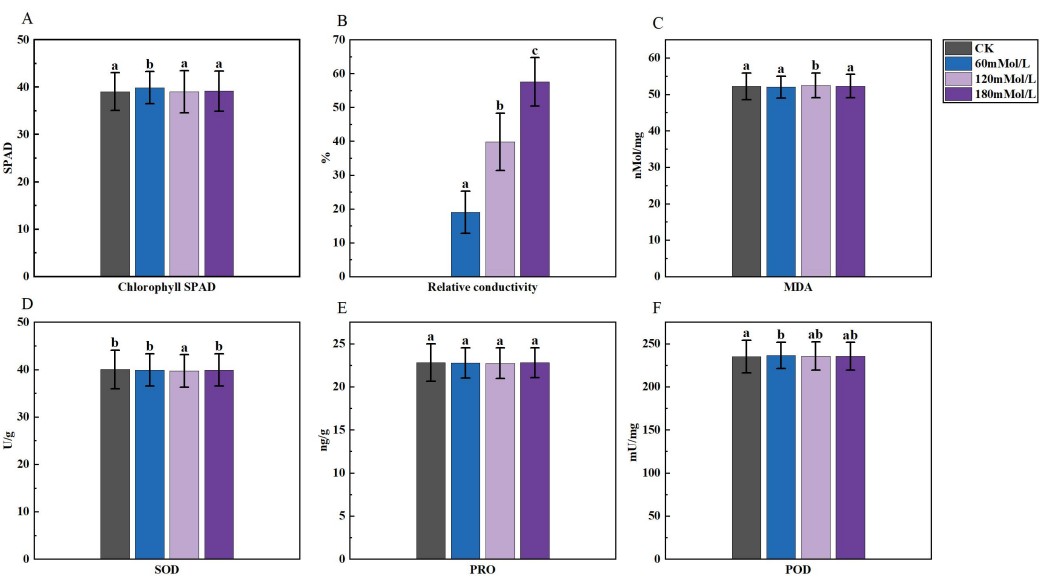

**Figure 2** **Analysis of difference in various physiological indicators under salinity stress.** (A–D) Markers of significance of differences at 0.05 level.

## RESULTS

### Analysis of difference among salt tolerance indicators

An analysis of variance (ANOVA) was conducted on the morphological, physiological, and germination indices of 41 maize varieties (Figs. 1–3). Plant height, stem thickness, germ length, radicle length, root crown ratio, leaf area, relative conductivity, chlorophyll SPAD, germination rate, water content, germination index, salt tolerance index, salt injury index, SOD, MDA, and POD were significantly different among the four NaCl concentration treatments, while PRO was not significantly different. The stem thickness, germ length, radicle length, leaf area, germination rate, germination index, salt tolerance index, and germination potential decreased with the increase of the salt concentration, while the conductivity and salt injury index increased. In this study, the POD of maize leaves differed significantly from the control group under 60 mMol NaCl treatment, but the changes were minimal as the NaCl concentration increased. The SOD activities in the functional leaves of 41 salt-tolerant maize varieties decreased and the MDA increased under 120 mM NaCl treatment. However, with higher concentrations of NaCl treatment, the SOD increased and the MDA decreased.

### Analysis of the correlation among the salt tolerance indicators

The correlation analysis of 18 indicators was carried out, and the relationship between the indicators is shown in Fig. 4. Plant height was significantly positively correlated with stem thickness and radicle length and negatively correlated with leaf area. Germ length was positively correlated with root-crown ratio, water content, germination index, and salt tolerance index, and negatively correlated with salt damage index. The four indicators, including root crown ratio, water content, germination index, and salt tolerance index,

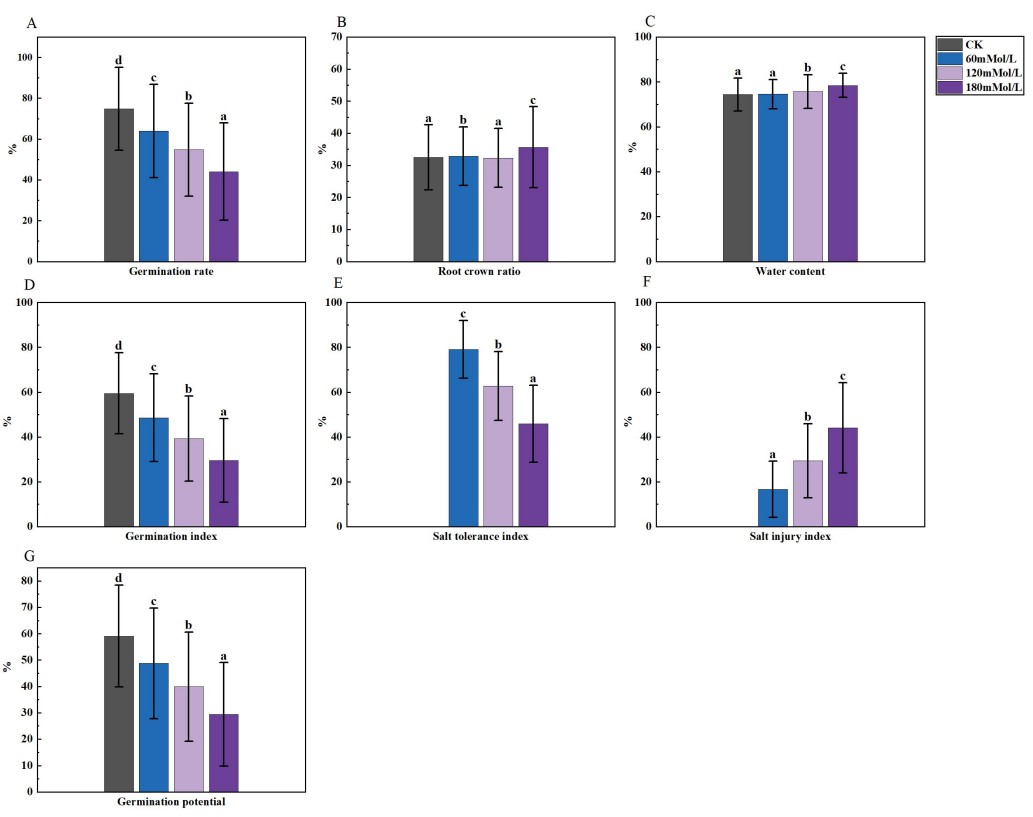

**Figure 3  Analysis of difference in various germination indicators under salt stress.** (A–D) Markers of significance of differences at 0.05 level.

were positively correlated with each other. Among the physiological indicators, chlorophyll SPAD was also significantly and positively correlated with MDA. MDA was negatively correlated with SOD. In addition, SOD was negatively correlated with POD. Among the germination indexes, germination rate was negatively correlated with germination potential. It was worth noting that the three indicators, root crown ratio, water content, and germination index, were all negatively correlated with the salt damage index.

## Principal component analysis and membership function analysis of salt tolerance indicator

Factor analyses were conducted on 18 indicators, as presented in Tables 2 and 3. Based on eigenvalues greater than 1, six principal components were extracted. The contribution rate of the six principal components extracted was 73.21%, and the distribution was relatively uniform, which could better summarize most of the indicators. The eigenvalue of the first principal component was 4.87. The germination potential had the highest loading value and was therefore referred to as the germination potential index factor. The eigenvalue of principal component 2 was 1.97, which was called the SOD factor. The plant height factor was assigned to principal component 3, which had an eigenvalue of 1.81. The germ length factor was assigned to principal component 4, which had an eigenvalue of 1.64.

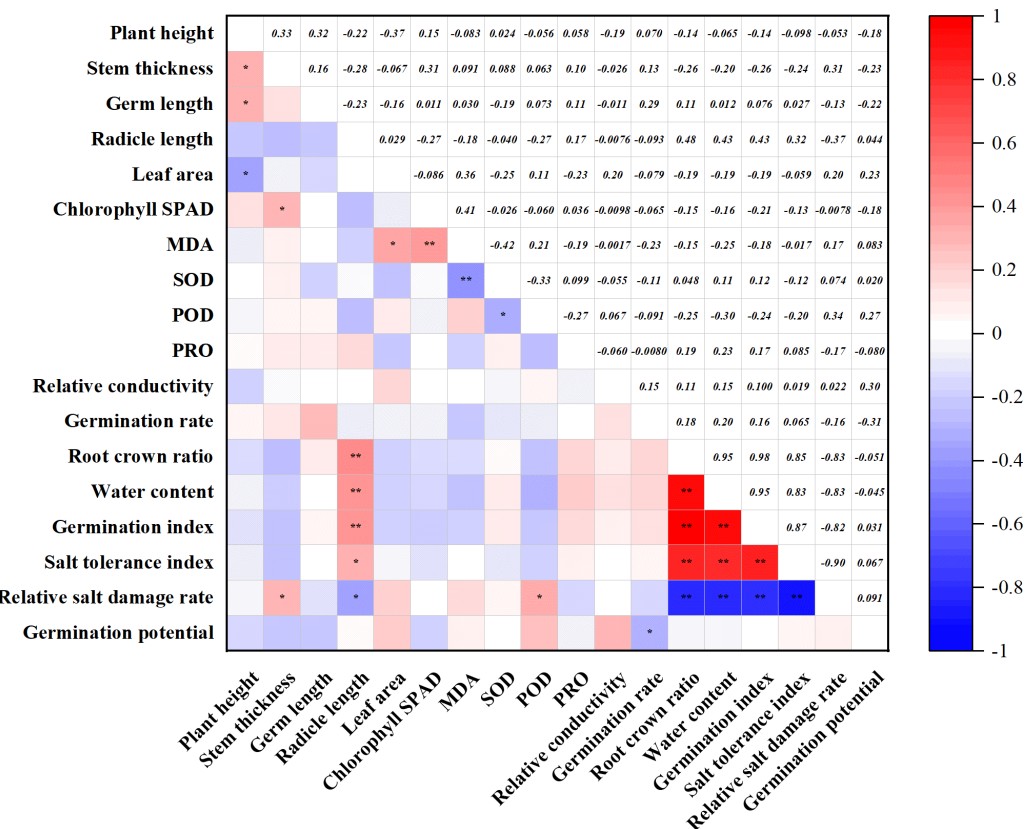

**Figure 4** Correlation of agronomic traits with physiological indicators. $*p \leq 0.05$ $**p \leq 0.01$.

Principal component 5, named the chlorophyll SPAD factor, possessed an eigenvalue of 1.57. Similarly, the sixth principal component, with an eigenvalue of 1.32, was associated with the highest loading value of PRO. The six factors obtained from the principal component analysis were comprehensively evaluated by a membership function, including the germination potential, POD, plant height, germination length, chlorophyll SPAD, and PRO. The D values of these factors were calculated using the membership function method, and the four NaCl concentrations listed in Table 4 were further calculated. In the absence of treatment, G33 exhibited the highest D value, while G18 had the highest D value under 60 mM NaCl concentration treatment. Conversely, G39 showed the highest D value in 120 mM NaCl concentration treatments. Meanwhile, G1 showed the highest D value in 180 mM NaCl concentration treatments. It is worth noting that the higher the membership function value, the stronger the salt tolerance. Therefore, among all cultivars, G18 displayed the highest salt tolerance in the 60 mM NaCl concentration treatment, G39 exhibited higher salt tolerance in the 120 mM NaCl concentration treatment, and G1 showed higher salt tolerance than other cultivars in the 180 mM NaCl concentration treatment.

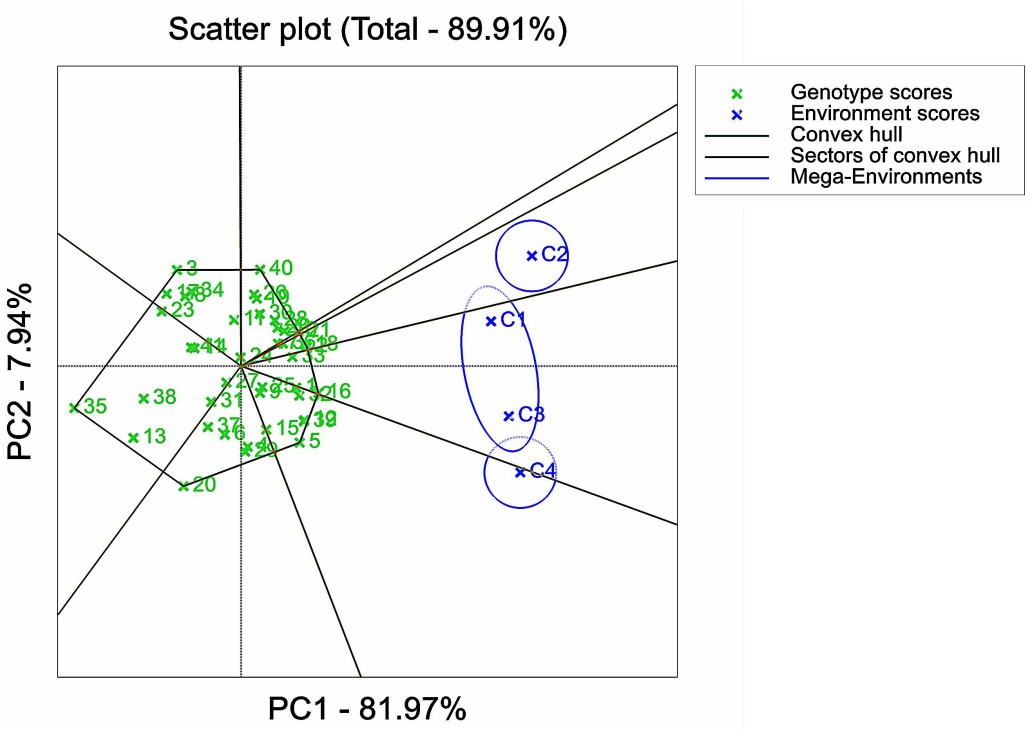

**Figure 5** **Adaptation analysis of maize varieties based on GGE biplot analysis.** 1, 2, 3, *etc.* are the Cultivar numbers of 41 maize varieties; C1, C2, C3 and C4 are the environments under stress of 0 mMol/L NaCl solution, 60 mMol/L NaCl solution, 120 mMol/L NaCl solution and 180 mMol/L NaCl solution, respectively.

## Salt tolerance or adaptation of different maize varieties based on the analysis of the GGE biplot

The D values of the test varieties under four stress concentrations were analyzed by the GGE biplot analysis. The first principal component accounted for 81.97% of the total variation, while the second principal component explained 7.94% of the variation. The GGE biplot was employed to visualize varietal adaptation functions, where the sectors represented the growth environments under the stress of the four NaCl concentrations. The maize varieties for trail numbers G3, G40, G21, G18, G16, G5, G20, and G35 were located in the apex of the sector area (Fig. 5). The polygon was divided into eight sectors, with the four environmental NaCl stress concentrations distributed in two sectors. C2 was located in the first sector, C4 was situated in the second sector along with the third sector, and C1 and C3 as a whole were mostly located in the second sector, but a small portion spanned the first sector. G18 exhibited closer proximity to the C2 region, suggesting its superior adaptation to the C2 environment compared to other varieties. On the other hand, G33 and G16 were located in the C1 and C3 regions, respectively; G12, G39, and G5 were closer to the C4 region, indicating that the above cultivars were better suited to the environment in their regions.

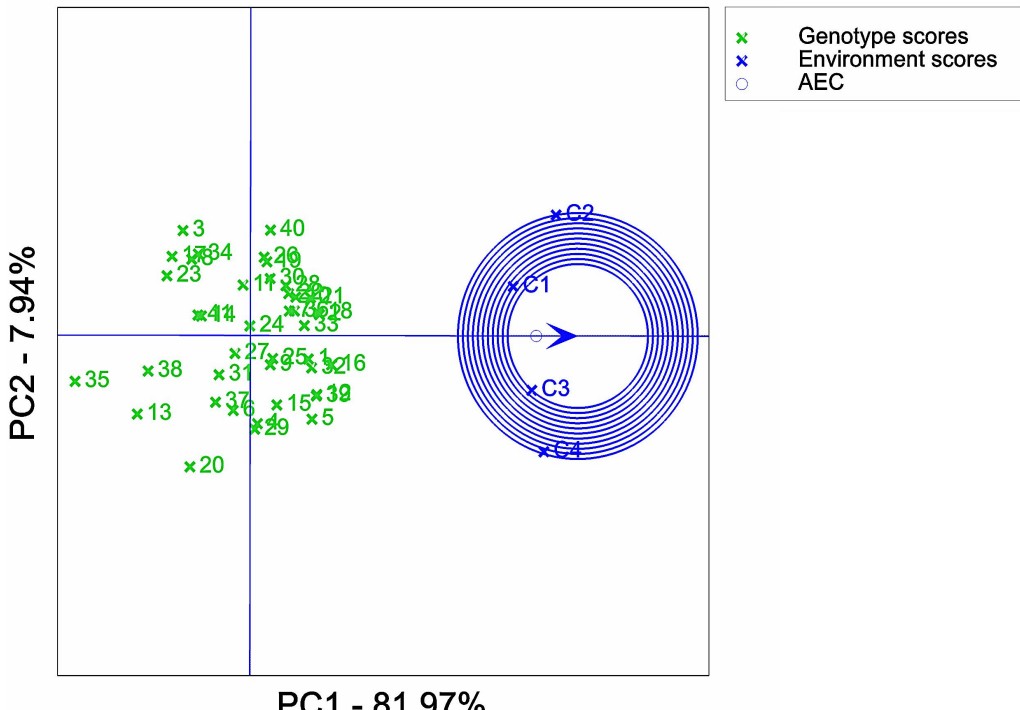

**Figure 6** **Analysis of optimal NaCl stress solutions based on GGE biplot.** 1, 2, 3, *etc.* are the cultivar numbers of 41 maize varieties; C1, C2, C3 and C4 are the environments under stress of 0 mMol/L NaCl solution, 60 mMol/L NaCl solution, 120 mMol/L NaCl solution and 180 mMol/L NaCl solution, respectively.

## Representativeness and discriminatory power of different NaCl concentrations

An important factor in assessing the suitability of the environment under NaCl stress concentration was its discriminatory power and representativeness. In the GGE biplot analysis, the circles connected the mean environmental axis and mean environmental value. The smaller the circle associated with a stress concentration environment point, the higher the overall level of that particular environment. It was evident that the comprehensive ranking of the four stress environments was C3 > C1 > C4 > C2 (Fig. 6), *i.e.*, 120 > 0 > 180 > 60 mMol/L. This implied that C3 exhibited a stronger ability to identify the salt tolerance and stability of maize germplasm, while C2 had the weakest ability to do so. It should be noted that C1 served as the control environment, which explained its limited capacity to identify the salt tolerance of maize but maintained high stability.

The environments under different stress concentrations were analyzed and depicted (Fig. 7). The angle between the environment under C1 stress concentration and the environments under C2, C3, and C4 stress concentrations was found to be less than 90°, indicating a strong positive correlation. This suggests that the ranking of varieties under the four stress concentrations was similar. In terms of representativeness, the environment

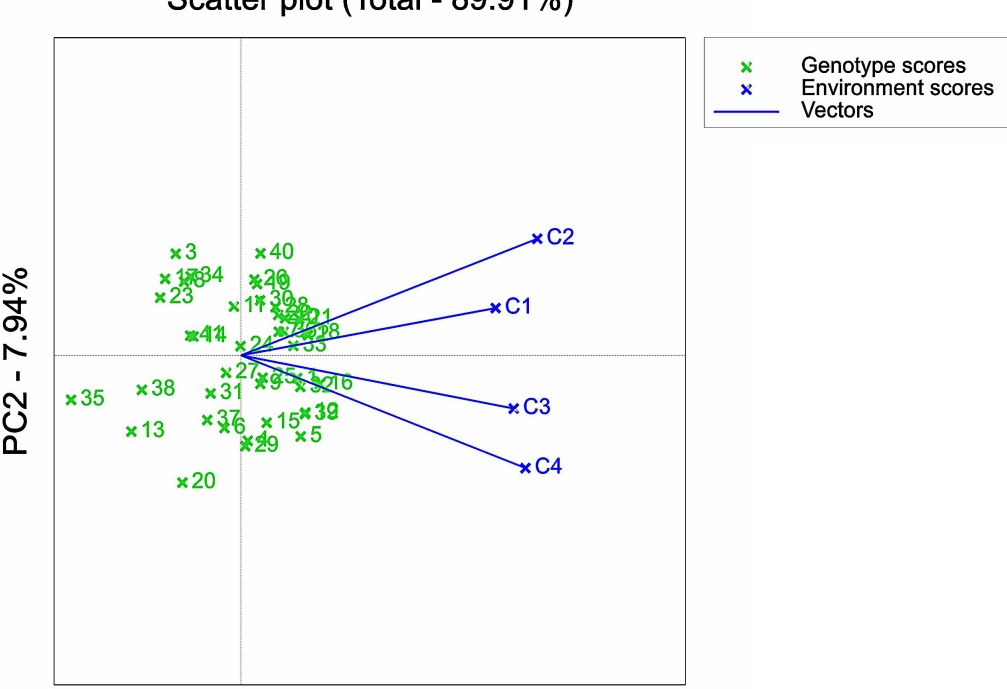

**Figure 7 Representative analysis of NaCl stressed solutions based on GGE biplot.** 1, 2, 3, *etc.* are the cultivar numbers of 41 maize varieties; C1, C2, C3 and C4 are the environments under stress of 0 mMol/L NaCl solution, 60 mMol/L NaCl solution, 120 mMol/L NaCl solution and 180 mMol/L NaCl solution.

under C1 stress concentration showed the best representation, as it had the smallest angle with the mean axis. On the other hand, environments C2 and C4 had the longest lines, indicating that in terms of the discrimination of individual varieties, the degree of variation of varieties in this environment was greater than that in other environments. However, this ignored the stability and salt tolerance of the environmentally discriminated cultivars in this environment.

Furthermore, the analysis of variance of the identification indexes of the experimental maize (Figs. 1–3) revealed a significant downward trend. There was also a significant distinction among multiple varieties and concentrations. These results suggested that the measured indexes at 120 mmol/L NaCl concentration could effectively identify the salt tolerance and stability of different maize varieties. Thus, this concentration could be considered an ideal environment for determining the salt tolerance of maize varieties, thereby providing strong persuasiveness and credibility. Moreover, the C1 environment indicated less discrimination among varieties, as it had the shortest line segment. Overall, the environments at the four stress concentrations could be divided into three regions: one including C2, another including C1 and C3, and one including C4.

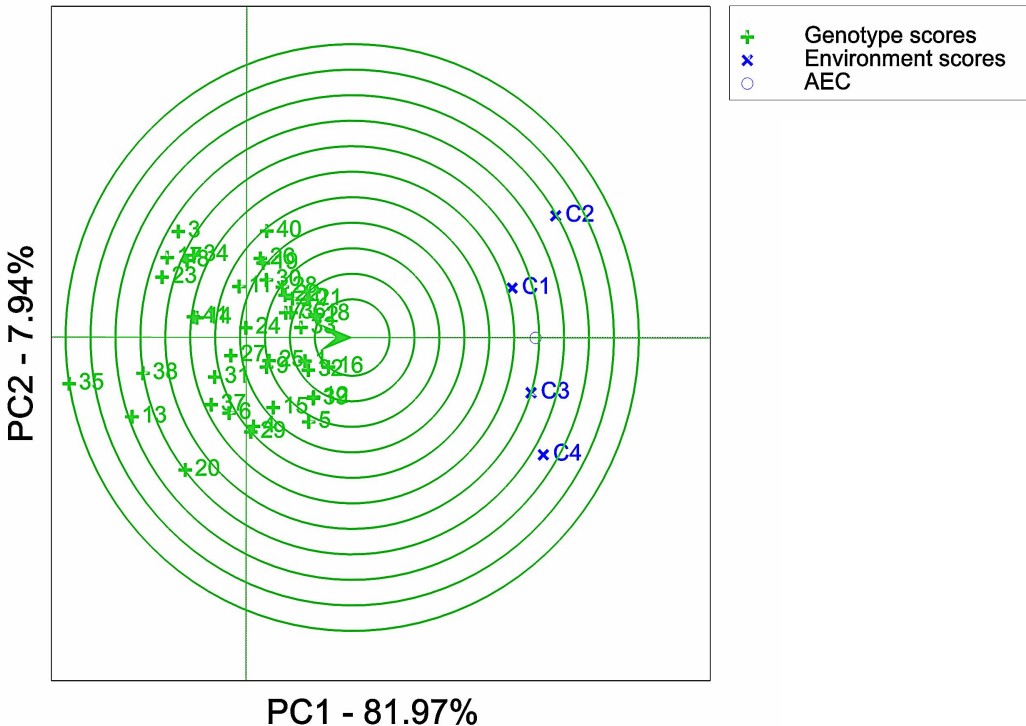

**Figure 8** **Varietal analysis of salt tolerance stability based on GGE biplot.** 1, 2, 3, *etc.* are the cultivar numbers of 41 maize cultivars; C1, C2, C3 and C4 are the environments under stress of 0 mMol/L NaCl solution, 60 mMol/L NaCl solution, 120 mMol/L NaCl solution and 180 mMol/L NaCl solution, respectively.

## Comparison of maize varieties

The center point of the circle on the environmental axis represents the average stress concentration. Combined with this experiment, the varieties closer to the center circle had the best overall performance in salt tolerance and stability. The top six varieties in terms of combined salt tolerance and stability were ranked as follows: G16 > G18 > G2 > G33 > G32 > G1 (Fig. 8), *i.e.,* Qun Ce 888 > You Qi 909 > Ping An 1523 > Xin Nong 008 > Xin Yu 66 > Hong Xing 990. On the other hand, the bottom six varieties in terms of combined salt tolerance and stability were ranked as follows: G35 < G13 < G38 < G20 < G3 < G23, *i.e.,* Feng Tian 14 < Xi Meng 668 < Ji Xing 218 < Gan Xin 2818 < Hu Xin 712 < HengYu 369. The variety ranking obtained by GGE biplot analysis was compared with the membership function value calculated by the membership function method. It was found that after conducting the GGE dual standard analysis of the D- values, the maize germplasm could be further comprehensively evaluated by combining stability and salt tolerance based on the D-value. This would help to identify germplasm resources with high potential for both stability and salt tolerance.

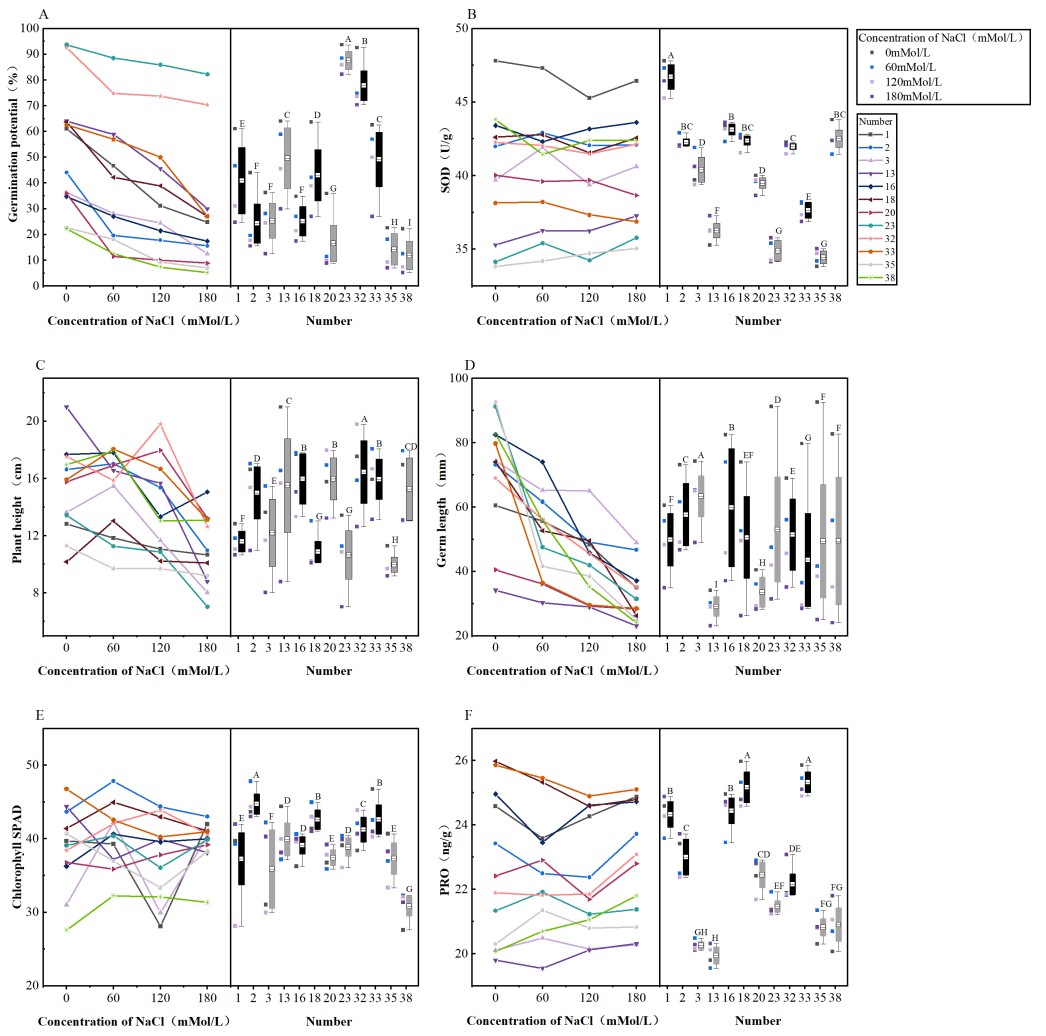

**Figure 9** (A–F) Differential changes in varieties under salt stress. Trend plots of a total of 12 cultivars with excellent and very poor salt tolerance and stability as a function of NaCl concentration (left); box plots of 12 cultivars under NaCl treatment (right); black plots indicate cultivars with excellent salt tolerance and stability, and grey box plots indicate cultivars with very poor salt tolerance and stability; numbers 1, 2, 3, 13, 16, 18, 20, 23, 32, 33, 35, 38 represent maize varieties Hong Xing 990, Ping An 1523, Hu Xin 712, Xi Meng 668, Qun Ce 888, You Qi 909, Gan Xin 2818, HengYu 369, Xin Yu 66, Xin Nong 008, Feng Tian 14, Ji Xing 218, respectively. The letters in boxplots are significance markers and confidence intervals at $P < 0.05$ are considered significant.

## Analysis of changes in selected cultivars with salt treatment

Indicators (germination potential, SOD, plant height, germ length, chlorophyll, SPAD, PRO) extracted from six principal component analyses of a total of 12 cultivars (six cultivars with the best salinity tolerance and stability, six cultivars with poor salinity tolerance and stability) were compared (Fig. 9). There were significant differences between treatments and cultivars in Figs. 9A–9E; the black box plots showed that the variation under the treatment was smaller than the grey box plots and performed better than the grey box plots in terms of the actual significance of the indicated indices. The germination potentials of

**Table 2  Maize trait contribution eigenvalues.** Extraction method: Principal component analysis.

| Factor | Initial eigenvalues | | | Sum of squared rotating loads | | |
|---|---|---|---|---|---|---|
| | Total | Percentage variance | Cumulative % | Total | Percentage variance | Cumulative % |
| 1 | 5.17 | 28.74 | 28.74 | 4.87 | 27.07 | 27.07 |
| 2 | 2.44 | 13.54 | 42.27 | 1.97 | 10.95 | 38.03 |
| 3 | 1.85 | 10.28 | 52.56 | 1.81 | 10.04 | 48.06 |
| 4 | 1.41 | 7.85 | 60.41 | 1.64 | 9.13 | 57.19 |
| 5 | 1.17 | 6.50 | 66.91 | 1.57 | 8.72 | 65.90 |
| 6 | 1.13 | 6.30 | 73.21 | 1.32 | 7.30 | 73.21 |
| 7 | 0.95 | 5.28 | 78.48 | | | |
| 8 | 0.73 | 4.03 | 82.52 | | | |
| 9 | 0.70 | 3.90 | 86.42 | | | |
| 10 | 0.67 | 3.73 | 90.15 | | | |
| 11 | 0.53 | 2.96 | 93.11 | | | |
| 12 | 0.40 | 2.24 | 95.35 | | | |
| 13 | 0.36 | 1.98 | 97.32 | | | |
| 14 | 0.31 | 1.70 | 99.02 | | | |
| 15 | 0.12 | 0.65 | 99.67 | | | |
| 16 | 0.04 | 0.20 | 99.87 | | | |
| 17 | 0.02 | 0.10 | 99.97 | | | |
| 18 | 0.01 | 0.03 | 100.00 | | | |

the 12 cultivars decreased with the increase of the salt concentration, and cultivars G23 and G32 showed better germination potentials than the other cultivars under salt stress (Fig. 9A). The activity of SOD also varied with salt stress concentration and the variation trend varied between cultivars with the increase of the concentration, but the graph clearly showed that variety No. 1 always maintained higher SOD activity than other cultivars (Fig. 9B). The plant height varied excessively between treatments, but when the NaCl concentration was 60 mMol/L and 180 mMol/L, the plant height of all varieties was more concentrated, and when the NaCl concentration was 120 mMol/L, the difference between varieties was large (Fig. 9C). The germ length of each variety became shorter with the increase of NaCl concentration, and in the folding line graph, the germ length of each variety became shorter when the NaCl concentration was 60 mMol/L; however, when the NaCl concentration was 120 mMol/L, the tendency of becoming shorter was not obvious compared with that of 60 mMol/L for some cultivars, and the germ length of some cultivars appeared to be significantly reduced. In the box plots, the dispersion of cultivars G23, G35, and G38 was large, indicating that several cultivars were less stable (Fig. 9D). The trends of chlorophyll SPAD changes with increasing NaCl concentration varied between cultivars; in the folded line graph, combined with the comparison of NaCl concentration at 60 mMol/L, 120 mMol/L, and 180 mMol/L, it was found that the degree of change was large when the concentration was 120 mMol/L. The variation trend of 60–120 mMol/L was different from 120–180 mMol/L, and the NaCl concentration of 120 mMol/L was the turning point. The chlorophyll SPAD among cultivars became more concentrated from a large degree of

**Table 3  Matrix of factor loadings after rotation.**

| Characters | Ingredients | | | | | |
|---|---|---|---|---|---|---|
| | 1 | 2 | 3 | 4 | 5 | 6 |
| Plant height | −0.070 | −0.087 | 0.747 | 0.122 | 0.056 | −0.176 |
| Stem thickness | −0.298 | −0.195 | 0.348 | 0.466 | 0.241 | 0.171 |
| Leaf area | 0.109 | 0.282 | 0.573 | 0.012 | 0.429 | 0.020 |
| Water content | 0.444 | −0.165 | −0.459 | −0.256 | −0.044 | −0.249 |
| Root-crown ratio | −0.165 | 0.407 | −0.638 | 0.120 | 0.022 | 0.198 |
| Germ length | −0.078 | −0.091 | 0.110 | 0.864 | −0.012 | −0.020 |
| Radicle length | −0.058 | 0.548 | −0.171 | 0.669 | −0.173 | −0.060 |
| MDA (nmol/mg) | −0.081 | −0.792 | 0.079 | −0.128 | −0.206 | 0.146 |
| SOD (U/g) | −0.247 | 0.635 | 0.254 | −0.138 | −0.258 | 0.233 |
| POD (mU/mg) | 0.189 | −0.449 | 0.103 | 0.102 | 0.062 | −0.099 |
| PRO | 0.084 | 0.031 | −0.181 | 0.035 | 0.074 | 0.858 |
| Chlorophyll SPAD | 0.110 | 0.028 | 0.108 | −0.143 | 0.826 | 0.245 |
| Germination potential | 0.957 | −0.100 | −0.034 | −0.087 | 0.085 | 0.059 |
| Germination rate | 0.927 | −0.213 | −0.017 | −0.096 | 0.085 | 0.113 |
| Germination index | 0.948 | −0.126 | −0.006 | −0.138 | 0.015 | 0.097 |
| Salt tolerance index | 0.938 | 0.086 | −0.025 | 0.007 | −0.055 | −0.016 |
| Salt injury index | −0.925 | 0.050 | −0.060 | −0.009 | −0.083 | 0.129 |
| Relative conductivity | 0.023 | 0.176 | −0.088 | −0.183 | −0.683 | 0.473 |

dispersion when passing through this concentration to higher concentrations, and then the differences were not significant (Fig. 9E). The PRO content showed an increasing trend at 120–180 mMol/L, where at 60–120 mMol/L, the PRO content of some varieties increased, and that of some varieties decreased, as shown in the NaCl concentration of 120 mMol/L served as a turning point for the screening index; in the box-and-line plots, the PRO content was less dispersed among the four treatments, but when the black box plots were compared with the grey box plots, the PRO content of the individual cultivars represented by the black box plots was higher than that of the grey box-and-line plots (Fig. 9F).

## DISCUSSION

### Effect of salt stress on morphological and physiological indexes of maize

Salinity stress is a significant abiotic factor that has a profound impact on maize yields in countries where maize is cultivated (Epstein et al., 1980). Salt stress causes plants to be affected by both excess ions and water deficits (Giambalvo et al., 2022). The detrimental effects of salt stress on plants include hindrance to key physiological processes, inhibition of growth and development, and disruption of cell structure (Frukh et al., 2020; Truong et al., 2018). Under NaCl stress, plants generate osmotic potential, which leads to a rapid reduction in cell expansion pressure, shrinkage of the plasma membrane, and ultimately growth inhibition (Munns & Tester, 2008). Previous research on salt tolerance in maize primarily focused on a single growth season, without considering multiple growth periods.

**Table 4 Combined D-values of germination potential, POD activity, SOD activity, plant height, germ length, chlorophyll SPAD, and PRO content at four Nacl concentrations.**

| Number | CK D-Value | 60 mMol/L D-Value | 120 mMol/L D-Value | 180 mMol/L D-Value |
|---|---|---|---|---|
| 1 | 3.42 | 3.16 | 2.60 | 3.29 |
| 2 | 3.38 | 3.36 | 3.11 | 2.94 |
| 3 | 2.16 | 2.87 | 1.91 | 1.92 |
| 4 | 2.88 | 2.52 | 3.00 | 2.65 |
| 5 | 3.44 | 2.82 | 3.37 | 2.99 |
| 6 | 2.64 | 2.46 | 2.72 | 2.53 |
| 7 | 3.16 | 3.11 | 3.10 | 2.57 |
| 8 | 2.48 | 2.60 | 2.29 | 1.79 |
| 9 | 3.18 | 2.74 | 2.80 | 2.70 |
| 10 | 3.41 | 3.08 | 3.12 | 2.54 |
| 11 | 2.76 | 3.03 | 2.26 | 2.54 |
| 12 | 3.14 | 3.15 | 3.39 | 3.03 |
| 13 | 2.12 | 1.60 | 2.29 | 1.65 |
| 14 | 2.83 | 2.27 | 2.52 | 1.87 |
| 15 | 3.22 | 2.62 | 3.02 | 2.76 |
| 16 | 3.34 | 3.36 | 3.20 | 3.22 |
| 17 | 2.52 | 2.30 | 2.39 | 1.43 |
| 18 | 3.29 | 3.41 | 3.09 | 2.95 |
| 19 | 3.40 | 2.91 | 2.72 | 2.31 |
| 20 | 2.34 | 1.95 | 2.46 | 2.38 |
| 21 | 3.17 | 3.37 | 3.36 | 2.63 |
| 22 | 3.28 | 3.16 | 2.77 | 2.71 |
| 23 | 2.65 | 2.18 | 1.99 | 1.68 |
| 24 | 2.93 | 2.79 | 2.64 | 2.46 |
| 25 | 3.06 | 2.83 | 2.97 | 2.62 |
| 26 | 3.02 | 3.10 | 2.84 | 2.26 |
| 27 | 2.75 | 2.61 | 2.67 | 2.37 |
| 28 | 3.22 | 3.09 | 3.19 | 2.36 |
| 29 | 2.56 | 2.70 | 2.87 | 2.80 |
| 30 | 3.21 | 2.99 | 2.84 | 2.37 |
| 31 | 2.47 | 2.48 | 2.78 | 2.21 |
| 32 | 3.22 | 3.14 | 3.29 | 2.92 |
| 33 | 3.57 | 3.02 | 3.03 | 2.78 |
| 34 | 2.98 | 2.41 | 2.20 | 1.81 |
| 35 | 1.93 | 1.22 | 1.35 | 1.40 |
| 36 | 3.18 | 3.22 | 2.87 | 2.80 |
| 37 | 2.49 | 2.43 | 2.36 | 2.55 |
| 38 | 2.10 | 1.97 | 1.98 | 1.88 |
| 39 | 3.29 | 3.03 | 3.45 | 2.95 |
| 40 | 3.04 | 3.27 | 2.80 | 2.28 |
| 41 | 2.62 | 2.34 | 2.60 | 1.81 |

Additionally, results may vary when diverse germplasms are utilized. For instance, *Fu & Zhang (2015)* discovered that a concentration of 100 mMol/L NaCl promoted maize seed germination, while treatments with 200 mMol/L NaCl and higher concentrations inhibited it. In our own experiment, various indicators related to seed germination, such as germ length, germination rate, and germination index, decreased with the increase of the salt concentration. Salinity disrupts the balance of nutrients and hormones during seed germination, particularly gibberellin/abscisic acid during germination, resulting in delayed germination (*Ullah et al., 2023*). Under osmotic stress, plants are capable of accumulating more solutes to maintain water content and alleviate osmotic stress caused by extracellular high osmolality (*Munns et al., 2020*). Contrary to the study conducted by *Xu et al. (2023)*, the relative water content did not decrease with the increase of the salt stress, but increased significantly at a concentration of 120 mMol/L NaCl. This increase may be attributed to the continuous salt stress, which induced swelling disorders in maize by reducing the water absorption efficiency (*Nandwal et al., 2020*).

In our study, we observed that the conductivity increased with the increase of the NaCl concentration. High salt concentration led to ionic toxicity, disruption of membrane systems, and increased membrane permeability, which further resulted in higher relative conductivity of the plant (*Esfandiari, Shamili & Homaei, 2020*; *Zhang et al., 2020*). Previous studies have found a significant increase in MDA levels (*Mittler, 2002*) and a decrease in SOD under salt stress, which may be attributed to the fact that NaCl stress reduced the activities of SOD and POD (*Gill & Tuteja, 2010*) and inadequately induced antioxidant systems (*Pan et al., 2018*). In our study, the physiological indicators of maize responded to salt stress. Under 120 mMol/L NaCl treatment, the activities of antioxidant enzymes SOD and POD in maize functional leaves decreased, while the levels of MDA increased. However, as the NaCl concentration increased, the levels of MDA decreased, SOD increased, and the changes in POD were not significant. Studies have shown that improving the activity of antioxidant enzymes and increasing antioxidant levels could prevent the excessive accumulation of reactive oxygen species (ROS) and alleviate oxidative damage induced by salt and alkaline stress (*Cao, Song & Zhang, 2022*). This is similar to a previous study in which the SOD and CAT activities of maples increased and then decreased under salt stress, while POD did not change much (*Xu et al., 2023*). Excessive NaCl impairs SOD metabolism by reducing and disrupting the production of important biomolecules and enzyme activities (*Allen, 1995*). Salinity induces the formation of ROS in plant cells, and excessive accumulation of ROS leads to oxidative damage to plant membrane lipids, proteins, DNA, and nucleic acids (*Neto et al., 2006*). The enzymatic protection system, consisting of antioxidant enzymes such as SOD, POD, CAT, ascorbate peroxidase (APOX), and glutathione reductase (GR) (*Kaur et al., 2014*), played a crucial role in scavenging reactive oxygen species and protecting against oxidative damage, thus maintaining plant metabolic homeostasis (*Khodary, 2004*; *Hossain, Mostofa & Fujita, 2013*), defense systems to scavenge high concentrations of ROS. Thus, higher salt concentration induces higher levels of enzymes in the antioxidant defence system of the plant. This explained the fact that in our study, compared with 120 mMol/L NaCl concentration, maize under 180 mMol/L NaCl concentration stress had higher SOD levels, with an increase in MDA.

## Analysing salt tolerance or adaptation of different maize varieties based on affiliation function method and GGE biplot

The predecessors mainly used the membership function method to evaluate the comprehensive performance of the sieve plant (*Liu et al., 2012*; *Liu et al., 2018*). Similarly, the affiliation function method has been employed for the comprehensive evaluation of crops, wherein the D-value can be calculated based on the weights and values of the affiliation function, subsequently facilitating the comprehensive evaluation of cultivars (*Rao et al., 1997*). However, the analysis of D-values using GGE biplots allows for the simultaneous evaluation of maize salt tolerance. This enables the determination of the stability of maize variety performance in different environments while assessing salt tolerance. In the present study, the GGE biplot analysis was utilized to analyze D values across four concentrations of NaCl concentrations. This approach enhanced the accuracy and reliability of the analyses, surpassing mere observations.

The GGE biplot analysis method encompassed the assessment of genotype and genotype-by-environment interactions (*Batista et al., 2017*). Through the use of scatterplots, this method visually represented bidirectional data, with genotypes as input and environment as output. The primary purpose of this method is to describe the super-environment, rank the genotypes, and identify stable environments (*Fu & Zhang, 2015*). In the GGE biplot polygon (Fig. 5), cultivars were positioned in specific environments, with the top representing the genotype. This positioning indicated that these particular genotypes exhibited superior performance in the environments in which they were located (*Yan et al., 2007*; *Saremirad & Taleghani, 2022*). Notably, the 120 mMol/LNaCl concentration was determined to be the optimal stress concentration for maize cultivars in this study. Based on the observation of the biplot in Fig. 6, it was deduced that the 120 mMol/LNaCl concentration, which was represented by the smallest circle, indicated the highest level of the respective environment (*Habtegebriel, 2022*). It is worth mentioning that in GGE biplots analysis, the angle among environment vectors reflects their correlation, and a smaller angle indicates a higher level of correlation (*Taleghani et al., 2023*). In contrast, the angles of the lines representing the environments at the four stress concentrations were less than 90° (Fig. 7), suggesting that most cultivars exhibited similar performance in these environments. The length of the environment vectors approximated the standard deviation within each environment and served as an indicator of environmental distinctiveness. Consequently, environments with a longer vector length had a larger standard deviation and were more distinguishable (*Saremirad & Taleghani, 2022*; *Yan & Kang, 2002*). Cultivars close to the center circle performed best (*Sharma et al., 2023*), and in combination with this experiment (Fig. 8), the maize cultivars Qun Ce 888, You Qi 909, and Ping An 1523 had the best stability and salt tolerance.

## Significance of screening salt-tolerant germplasm resources for future cultivation and development of salt-tolerant crops

Soil salinization had become a prominent problem hindering ecological development and economic progress, and soil salinity and drought were considered to be the most important environmental stresses worldwide due to their negative effects on plant growth and crop
productivity (*Singh et al., 2022*). It has been projected that approximately half of the world's arable land will be affected by salinity by the year 2050 (*Butcher et al., 2016*). In response to this challenge, Fita et al. have proposed the development of drought and salt stress, aiming to significantly increase food production in the near future (*Fita et al., 2015*). A potential solution lies in the cultivars in mudflat saline lands, with the establishment of thresholds to control salinity levels within acceptable limits (*Zhang et al., 2022*). The development of saline agriculture was of great practical significance to China's food industry, ensuring an efficient food supply and maintaining the ecological balance of the region (*Kong et al., 2019*). Therefore, cultivating salt-alkali-tolerant plant varieties and enhancing the salt-alkali tolerance of plants proved to be an effective biological measure to alleviate the impact of saline-alkali land on plants. It also generated favorable ecological and economic benefits and promoted the sustainable development of agriculture (*Wang et al., 2017b*). In many countries,saline-alkali land has been improved and its potential developed through salt-tolerant plant experiments and the cultivation of salt-tolerant plants. At that time, over 100 countries grappled with saline soils, including Chinese major grain-producing areas in the northwest, north, and northeast, where grain production and quality were affected. Therefore, apart from soil improvement measures, it is imperative to cultivate salt-tolerant crop varieties, because this can increase maize yields in saline soils across China (*Ye et al., 2019*). For example, some people improved crop yields in saline soils by regulating inter-root microorganisms (*Ren et al., 2023*), while others improved saline soil properties by applying rice straw, *etc.* (*Shaaban et al., 2023*), and still others enhanced saline soil properties by applying sand (*Wang et al., 2023*). Therefore, the comprehensive evaluation of salt tolerance in maize cultivars and the screening of salt-tolerant cultivars by an integrated approach could provide a basis for breeding more superior cultivars and have far-reaching significance for the selection of salt-tolerant germplasm suitable for the application of saline soil.

## CONCLUSIONS

The morphological and physiological indicators of 41 different maize varieties at the germination stage and seedling stage were analyzed by principal component analysis (PCA), subordinate function method (SFM), and GGE biplot analysis (DLAS) to evaluate and screen six maize varieties with stable salt tolerance (Qun Ce 888, You Qi 909, Ping An 1523, Xin Nong 008, Xinyu 66, and Hong Xin 990) and six maize varieties with poor stable salt tolerance (Feng Tian 14, Xi Meng 668, Ji Xing 218, Gan Xin 2818, Hu Xin 712, and Heng Yu 369). The response of maize to salt stress was studied. The morphological indicators of maize decreased with the increase of salt concentration, and MDA, SOD, and POD changed under the stress of 120 mMol/L and 180 mMol/L NaCl, indicating that salt stress changed the amount of antioxidant enzymes in the body at this concentration and formed a certain salt tolerance mechanism. The research results provided a theoretical basis for breeding and promotion of maize genotypes in saline regions. Additionally, the GGE double standard analysis was used in the experiment to screen out the optimal stress concentration while screening out the comprehensive performance of each variety under

salt stress, which provided a more accurate evaluation method for the salt tolerance of varieties.

Such research endeavors will further advance our understanding of maize cultivation in challenging environments and pave the way for sustainable agriculture practices.

## ACKNOWLEDGEMENTS

The authors are grateful to the anonymous reviewers for their valuable comments and suggestions.

### Funding
This work was funded by the National Natural Science Foundation of China, grant number 31760388. The funders had no role in study design, data collection and analysis, decision to publish, or preparation of the manuscript.

### Grant Disclosures
The following grant information was disclosed by the authors:
The National Natural Science Foundation of China: 31760388.

### Competing Interests
The authors declare there are no competing interests.

### Author Contributions
- Huijuan Tian conceived and designed the experiments, performed the experiments, analyzed the data, authored or reviewed drafts of the article, and approved the final draft.
- Hong Liu conceived and designed the experiments, performed the experiments, analyzed the data, prepared figures and/or tables, and approved the final draft.
- Dan Zhang conceived and designed the experiments, performed the experiments, analyzed the data, prepared figures and/or tables, authored or reviewed drafts of the article, and approved the final draft.
- Mengting Hu performed the experiments, prepared figures and/or tables, and approved the final draft.
- Fulai Zhang performed the experiments, prepared figures and/or tables, authored or reviewed drafts of the article, and approved the final draft.
- Shuqi Ding performed the experiments, prepared figures and/or tables, and approved the final draft.
- Kaizhi Yang performed the experiments, authored or reviewed drafts of the article, and approved the final draft.

### Data Availability
Raw measurements can be found in Files S1 and S2. The data of these phenotypic and physiological indices were used for statistical analyses to compare their differences, to make correlation analyses, principal component analyses, affiliation function analyses, and GGE biscriterion analyses to make a screening of different maize varieties for salt tolerance.

## Supplemental Information

Supplemental information for this article can be found online at http://dx.doi.org/10.7717/peerj.16838#supplemental-information.

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
