# Peer review of "Screening of salt tolerance of maize (Zea mays L.) lines using membership function value and GGE biplot analysis"

_PeerJ, doi:10.7717/peerj.16838_

## Round 0.1 · original submission · Major Revisions

Are you sure that you prepared the manuscript according to our journal's author instructions? Please read carefully it and edit accordingly.

The language of the manuscript is very bad. I think you should have it edited by a fluent English speaker. I understand what you are saying, but some sentences seem incomplete. Also, there are many grammatical errors.

The introduction is very poorly written. Disconnected information is given, far from a flow. In-text reference acknowledgments are given inappropriately. It's written quite clumsily. You did not fully explain the purpose of the study. You have to rearrange everything.

Line 47: Zea mays shold be written in italics
Line 65-66: radicle length and radicle length?????? correct please
Are the germination tests you mentioned in the materials and methods section your discovery? If not, you need to cite the author.
What does it mean that MDA, PRO content, SOD, POD activity analyzes were performed according to the kit method? State clearly which kit you used.
What does 'Calculation formula' mean? It is not clear what you are calculating. You're explaining everything in a confusing way. Are these formulas your discovery? Why is there no reference to authors?
Line 202: Factorial analysis? or Factor analysis???
Why did you choose 8 factors? Do you think factors with an Eigen value below 1 are appropriate?
Line 281: what do you mean? "tolerant germplasm resources with high potential. stability."??????
If you seperated the Discussion section, why did you write the discussion sentences in the Results section? The presentation of the findings is very poor.
Line 302-303: I think you should reference what the previous results were here.
Line 358-375: Are these sentences appropriate for the discussion section? Or should it be in the introduction section?
You have not adequately discussed your findings.
The figures in the manuscript should be improved. It's very badly drawn.
Is the salt resistance of the material you use known? Did you test it for the first time? Not using a control cultivar (sensitive and tolerant) is a big shortcoming.
Why was the D value in Table 4 calculated only for SPAD?

Your manuscript looks very complicated. Taking into account the shortcomings noted by reviewers, my decision is a major revision.

However, I would like you to know that if you cannot correct it to the desired level, my second decision will be "reject".

You did a good job using GGE biplot analysis, but your explanation of it was inadequate. You should also explain other analyses better. The manuscript deserves a chance.

**Language Note:** The Academic Editor has identified that the English language must be improved. PeerJ can provide language editing services - please contact us at copyediting@peerj.com for pricing (be sure to provide your manuscript number and title). Alternatively, you should make your own arrangements to improve the language quality and provide details in your response letter. – PeerJ Staff

Reviewer 1 ·

Basic reporting

'no comment'

Experimental design

In materials and methods you did not mention the type of statistical design used

Validity of the findings

no comment

Additional comments

Manuscript Title: Screening of salt-tolerance of maize varieties based on the value of the membership function and under GGE biplot analysis
This research study examined seedling responses of 41 maize varieties under four salinity levels of 0, 60, 120, and 180 mM NaCl were grown and were applied 7 days of treatment.
I have found some weaknesses in the manuscript:
Minor comments
-In materials and methods you did not mention the type of statistical design used.
- in Figures, analysis of variance a shows high experiment error.
major comments
- At least should use a tolerant cultivar or sensitive to the salinity as a check.
- Seven days is very little time for accurate judgment on cultivars tolerance of salinity morphologically.
- GGE biplot analysis was designed for the Study of environmental interactions

·

Basic reporting

- The introduction's content is very poor. In the introductory part, authors are asked to present information on the most recent prior study, restrictions (why do this sort of research? ), the justification of your investigation, and the expected output that will be useful on the global research aspect.

- In "material and method", need information, average climate data during the study period, layout, and design of the study. Also, request to mention the procedures/measures of data collection with valid reference in material and method.

- "result and discussion": very immature writeup. the interpretation is not countered by recent published work. Need to include more valid citation. In discussion must need to add more recent references to improve the interpretation quality. In the discussion, please add recent references.

- In conclusion, the author must focus on the exact output and recommendations for future researchers.
Manuscripts need English editing to improve the text quality.

-References need to be checked. Some references are missing in the text of the manuscript.

Overall, the manuscript can be accepted after revising the above-mentioned points properly.

Experimental design

no comments

Validity of the findings

no comments

Additional comments

Plese revise the manuscript as per highlighted comments throughout the manuscript.

Reviewer 3 ·

Basic reporting

The present manuscript is sufficiently well written with proper use of professional English throughout the manuscript. The cited literature is appropriate and justifies the findings presented. Figures presented are of satisfactory quality however can be improved and raw data has also been shared which helped in understanding the content presented.

Experimental design

Experimental design presented in the manuscript is apt for the intended experiment and the materials and methods section is pretty self explanatory for anyone interested in replicating the experiment. The authors has done a commendable job in preparing the materials and methods as the details provided are appreciated.

Validity of the findings

The experiment presented is of immense importance in the present scenario, the results I believe have sufficient novelty for publication, however, the presentation of results is highly confusing as upto materials and methods the article suggests to present data for identification of salt tolerant genotype but when we reach result section authors have compared treatments only without the mention of genotype performance. For example in the plant height graph, are the heights of 41 genotypes averaged? if yes, then what was the basis for that and if not where do you address the varietal variation of plant height as is with the other parameters.
the results section needs major restructuring, apart from comparing salt treatments graphs for varietal variation must be included which will help in understanding the conclusion where in lines 378-386 it has been mentioned that 5 maize genotypes have been identified.

Additional comments

GGE bioplot and correlation analysis are self explanatory and figure quality is quite good. But authors need to revise the manuscript to show the varietal variation of the parameters measured which is the main crux of the article presented.

---

## Round 0.2 · Major Revisions

Dear authors
I appreciate your effort to improve your manuscript based on editorial and reviewer comments. However, there are still many points that need to be corrected. Even if the reviewers find your corrections sufficient, please review the corrections I am requesting below, and resubmit.

Abstract
Line 33-35: Please revise the sentence as “Principal component analysis identified 6 major components including germination vigor, peroxidase (POD), plant height, embryo length, SPAD chlorophyll and proline (PRO) factors.”
Material and Methods
Line 125-129: Please revise the sentence as “The germ length and radicle length were measured. In addition, the relative germination potential, relative germination rate, salt tolerance index, and salt injury index were calculated. The mentioned measurements and calculations were made with reference to Liu et al [36].”
Line 131: Add title. It could be "Physiological seedling measurements in outdoor experiment".
Line 140: Please delete “H T” When giving references in the text, only the surname of the first author is written. The initials of his name are not written. Check and correct throughout the text.
Line 154: “Ghoulam et al [38]” not “Ghoulam, C et al” When giving references in the text, only the surname of the first author is written. The initials of his name are not written. Check and correct throughout the text.
Line 156-158: It was unnecessary and very inexperienced for you to write them this way. After writing the name of the measurement you made on the top line, you can cite it in parentheses. For example “ Then the dry weight of the biomass was weighed, and the root-crown ratio (root-crown ratio=aboveground dry weight/underground dry weight) was calculated [13]. Follow this method throughout the entire text. Do not write again and again that we "refering" it to this and that.
Line 167: “POR” or “PRO”???
Line 172: Please revise as “Calculation formula of identification index”
Line 191-194: Please revise as “IBM SPSS Statistics 25 software was used to analyze the variance of examined traits of 41 maize varieties.” In the following lines, instead of writing the names of all the traits one by one, use the term "examined traits.
Results
Line 212: Please revise as “Analysis of difference among salt tolerance indicators”
Line 214: Please delete “refer to” unnecessary word
Line 223: Please revise to “SOD of antioxidative enzymes” as “SOD activities”
Line 228-230 “The correlation analysis of 18 indicators was carried out, and the relationship between the indicators is shown in Figure 4.” and “Correlation analyses were carried out on 18 indicators, and the relationship between these indicators is shown in Figure 4.” are mentioned same thing. Please delete one of them.
Line 230-232: “not only”……..”but also” is not appropriate conjunction. Please revise.
Line 242: Please change to “index” as “indicator”
Line 244: “Factor analyses” not “Factorial”. Please revise
Line 254: Please delete “called the PRO”
Line 268-313: What do the things you coded as C1, C2, C3 and C4 mean? Probably salt doses but you've never said that before. Please indicate which one corresponds to which. Or write it as it is, not like this.
Line 284: Please revise to “concentrations of NaCl” as “NaCl concentrations”
Line 314: Please revise as “comparison of maize varieties”
Line 327-329: Are you sure? I think this sentence is wrong. You didn't do the principal component analysis only for the 6 genotypes you found poorest. As a result of this analysis, you identified them. Revise the sentence so it is understandable.
Line 330: Please revise to Figure A-E as Figure 9A-E
Line 356: Please write the first letter of “The” word in lowercase
Line 387-410: This part is explained in a complicated way. Give the discussion about MDA first and then explain the enzymes. You get repetitive in the beginning and last sentences of this section. Rearrange it by doing it in a logical order.
Line 479-481: You gave the codes of these varieties in the previous lines, now you wrote their names. Write your names in the places you mentioned before.
Line 482: Change to “indexes” as “indicators”
Make same the subheadings in the "Results" and "Discussion" sections.
Do not use different expressions such as "NaCl concentration" in some places, "NaCl solution" in other places, etc., choose one throughout the text and make them all uniform.
Figure titles and what is written under the figures are not the same. For example, under Figure 1, you rewrote the same title as Figure 5. Remove these by arranging the Figures.
Also write down which varieties the numbers in Figure 2 correspond to.
Please delete this sentence “A: Analysis of the difference in chlorophyll content among four NaCl solutions; Analysis of the difference in conductivity among four NaCl solutions; Analysis of the difference in MDA content among four NaCl solutions; Analysis of the difference in SOD activity among four NaCl solutions; Analysis of the difference in PRO content among four NaCl solutions; Analysis of the difference in POD activity among four NaCl solutions;” in Figure 3 heading. Unnecessary
Please delete this sentence “A : Analysis of the difference in germination percentage among four NaCl solutions; B:Analysis of the difference in root crown ratio among four NaCl solutions; C:Analysis of the difference in water content ratio among four NaCl solutions; D:Analysis of the difference in germination index among four NaCl solutions;E: Analysis of the difference in salt tolerance index among four NaCl solutions; F : Analysis of the difference in salt damage index among four NaCl solutions; G:Analysis of the difference in germination potential among four NaCl solutions;” in Figure 6 heading.
Please delete this sentence “A: Analysis of the difference in plant height among four NaCl solutions; B: Analysis of the difference in stem thickness among four NaCl solutions; C: Analysis of the difference in germ length among four NaCl solutions; D: Analysis of the difference in radicle length among four NaCl solutions; E: Analysis of the difference in leaf area among four NaCl solutions;” in Figure 8 heading.

·

Basic reporting

Now need a minor modification on Title. It should be " Screening of Salt tolerance of Maize (Zea mays L) Lines using Membership Function Value and GGE biplot analysis"

Experimental design

Experimental design is appropiate

Validity of the findings

The findings are valid.

Additional comments

The authors have revised the manuscript to address all comments and suggestions.
Now need a minor modification on Title. It should be " Screening of Salt tolerance of Maize (Zea mays L) Lines using Membership Function Value and GGE biplot analysis"

Reviewer 3 ·

Basic reporting

The improvements suggested to the authors have all been incorporated and i am satisfied with it

Experimental design

The presented research do come under the scope of the journal and also has sufficient novelty for publication.

Validity of the findings

The authors have explained their concept behind the preparation of graphs and presentation of results, and i am sufficiently satisfied with the presented arguments and accept the same.

Additional comments

no comments

---

## Round 0.3 · accepted · Accept

Your manuscript is accepted after your final revisions.